# Directed graph transformers meet metabolic networks

## Abstract

Technical advances in sequencing have allowed the reconstruction of genome-scale metabolic models (GEMs) for a wide range of microorganisms. These models have been particularly useful for the prediction of essential genes and reactions, which are potential targets for antimicrobial therapies. However, current methods for essentiality prediction are computationally limited and are not able to accommodate the increasingly available data. Motivated by the success of data-driven approaches in other domains, this work introduces the metabolic transformer, a model designed for holistic identification of essential reactions in genome-scale models, entirely trained on synthetic knock-out data. It is demonstrated that the problem of essential reaction prediction can be theoretically formulated as the identification of redundant nodes in directed bipartite graphs. This reveals the limitations of message-passing schemes and motivates the development of a novel graph transformer architecture specifically tailored for metabolic networks. The proposed architecture is capable of addressing the essential reaction identification problem by capturing both the directionality and global structure of metabolic networks. To demonstrate the effectiveness of our approach, we composed a large-scale dataset of genome-scale models reconstructed from real microorganisms. [1].

## 1 Introduction

In recent years, the emergence of high-throughput technologies allowed the integration of transcriptomic data of multiple pathogens into large biological datasets. This integration paved the way for the reconstruction of microorganism metabolism which directly led to the possibility of modelling these systems computationally Ric (2020).

Metabolism is the set of basic life processes that take place in the cell. All the metabolic chemical reactions that occur in a cell form a metabolic network. Genome-Scale models (GEM) are structured biochemical, genetic, and genomic databases for an organism, that aim to cover the whole metabolic network of a cell. As of 2019, more than 6.000 GEMs have been reconstructed for organisms including bacteria, archaea and eukaryotes and over 140.000 automated reconstructions are available from over 2.600 organisms Büchel et al. (2013); Gu et al. (2019); Monk et al. (2017).

GEMs have been particularly useful for phenotype simulation and prediction. These models have shown great accuracy in predicting the growth rate of microorganisms along with the flux carried out for each reaction of metabolism Orth et al. (2010). This has been particularly interesting for the identification of essential reactions and genes. Essential reactions are those that are required for the growth of the organism Henry et al. (2006). Consequently, when these reactions are knocked out the growth in the model is null, i.e., the organism dies. The genes that encode these reactions are appealing therapeutical targets.

Although the results on essentiality prediction are promising, current methods can be computationally demanding and their application to uncertain data can be challenging. Improving the accuracy of predictions is not trivial and requires an increasing complexity on the models that sometimes involves an exponential combinatorial cost or requires the estimation of missing data O'brien et al. (2013); Salvy & Hatzimanikatis (2020). In addition, recent advances in metabolism profiling technology Judge et al. (2019) will produce large amounts of data that could be hard to accommodate into current

---

[1]The dataset and source code will be made publicly available in the final version.

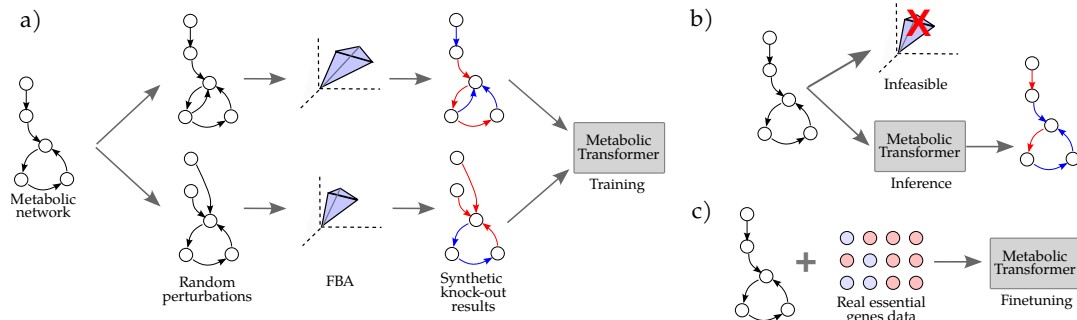

Figure 1: The metabolic transformer. *a)* The model is trained entirely from synthetic knock-out data obtained from metabolic network reconstructions from different microorganisms. *b)* Once trained, the model enables the computation of essential reactions in models where synthetic knock-out computations were previously infeasible. *c)* The model trained on syntehtic data can serve as a backbone for finetuning with real data.

modelling algorithms. In this context, there is a growing belief that data-driven methods (i.e. deep learning) have the potential to improve metabolic modelling Zampieri et al. (2019).

In this work, we propose a supervised learning approach, fully trained on synthetic knock-out data, for the holistic prediction of essential reactions in genome-scale models using graph neural networks and transformers models. Graph Neural Networks (GNN), first proposed in Scarselli et al. (2008), have become the reference algorithm for learning on graph-structured data. It is well known, however, that GNNs suffer from over-squashing and fail to model long-range relationships in graphs, which is a fundamental requirement for metabolic modelling. Motivated by the success of transformers models in natural language processing and computer vision, many works have proposed the use of transformers for graph-structured data Min et al. (2022); Rampášek et al. (2022); Bo et al. (2023). One of the early applications of transformers to graphs was presented in Dwivedi & Bresson (2020), where the authors proposed a generalization of transformers for graphs by applying the attention mechanism to the neighbourhood of each node. Later examples such as Kreuzer et al. (2021) and Kim et al. (2022) propose approaches where attention is applied to the whole graph. Other works have also proposed the integration transformers with GNNs to capture both the local and global information of the graph Rong et al. (2020); Rampášek et al. (2022).

This work proposes the metabolic transformer, a graph transformer model that combines directed message passing with positional encodings for essential reaction prediction in genome-scale models. Previous models have been proposed before for essentiality computation on GEMs Hasibi et al. (2024). These methods however rely only on message-passing schemes or are narrowed down to a specific dataset, which makes it application to other tasks or microorganisms limited. In this work, a model is proposed that is able to generalize essentiality reaction prediction across different microorganisms by pre-training on synthetic knock-out data. It is shown that the model can serve for effective transfer learning to real metabolic data, in this case, gene essentiality obtained experimentally in the lab. This is in contrast with existing LP methods that cannot generalize outside the modelled problem. To the best of our knowledge, this is the first model to successfully demonstrate transfer learning on metabolic data.

In summary, our proposed approach has the following properties:

- For inference, the proposed method only takes as input the topology of the metabolic network. This is in contrast with previous methods that require precomputing synthetic knock-out data for inference Freischem et al. (2022); Hasibi et al. (2024).

- The method predicts the essentiality of all reactions considering the whole network at once. This property is a direct consequence of the graph transformer architecture, which can take the features of all nodes as input. This overcomes the need for GNN-based approaches that require pre-computing global features on the network Yang et al. (2022).

- Given the high throughput of deep learning models, we are able to compute predictions for all reactions from only one forward pass of the model. This is an improvement compared

with the traditional synthetic knock-out simulation, which requires the computation of the growth rate for each reaction individually Oyelade et al. (2018).

- Metabolic data available for many microorganisms is scarce or limited. Aditionally, metabolic modelling is not limited to essentiality prediction, but also includes a wide range of other modelling tasks such as flux prediction, control, or drug target identification. By training a model on synthetic data from many diverse microorganisms, we developed a model that can serve as backbone for effective transfer learning on different organisms or tasks, where data can be limited. We thus believe that this is a significant step towards the creation of foundational models of metabolism.

Currently, the majority of GEMs publicly available are built though automated reconstruction pipelines Gu et al. (2019). As a consequence, most of these models lack manual curation and and, hence, traditional methods for knock-out simulations based on linear programming cannot be applied. In this context, since the proposed method only depends on the topology of the network, and does not require any additional data to be curated, it has the potential to enable the prediction of essential reactions in a wide range of models for which current methods are unable to produce predictions.

In addition to the proposed methods, we are releasing a carefully curated dataset of 23k GEMs from more than 100 different microorganisms. The dataset consists of large-scale graphs with an average of 2k nodes. This is the first dataset of GEMs and, given its dimensions, we believe it is of interest for the graph-learning community, making it one of the largest datasets in terms of number of nodes Hu et al. (2021).

Although this work is particularly focused on genome-scale metabolic models, the architecture and methods proposed in this work are not limited and can be applied to other topics traditionally modelled with directed bipartite graphs, such as control and distributed systems Murata (1989); Heiner et al. (2008), chemical reaction networks Wen et al. (2023) or retrosynthesis Chen et al. (2020).

The rest of the paper is organised as follows. In Section 2 preliminary definitions related to constraint-based models are introduced. Section 3 introduces the background for graph transformers used in this work. Section 4 describes the proposed architecture, and Section 5 describes the results obtained in the created dataset. Finally, Section 6 concludes the paper.

## 2 PRELIMINARY DEFINITIONS

In this Section, we introduce the preliminary definitions that are used throughout the paper. In particular, we introduce the formal definition of constraint-based models, flux balance analysis, and essential reactions.

### 2.1 CONSTRAINT-BASED MODELS

A *constraint-based model* Varma & Palsson (1994); Orth et al. (2011) is a tuple $\{\mathcal{R}, \mathcal{M}, \mathcal{S}, L, U\}$ where $\mathcal{R}$ is a set of *reactions*, $\mathcal{M}$ is a set of *metabolites*, $\mathcal{S} \in \mathbb{R}^{|\mathcal{M}| \times |\mathcal{R}|}$ is the stoichiometric matrix, and $L, U \in \mathbb{R}^{|\mathcal{R}|}$ are *lower and upper flux bounds* of the reactions.

$$2H_2 + O_2 \rightarrow 2H_2O$$

(a) Chemical reaction.      (b) Petri net.

Figure 2: A constraint-based model consisting only of the reaction in (a), can be modelled by the Petri net of (b).

All the reactions of the model are associated with a set of reactant metabolites and a set of product metabolites. For example, the reaction $r{:}A \rightarrow 2B$ has a reactant metabolite $A$, and a product metabolite $B$, with stoichiometric weights 1 and 2 respectively, i.e. reaction $r$ consumes one molecule of type $A$ and produces two molecules of type $B$. The stoichiometric matrix $\mathcal{S}$ accounts for all the stoichiometric weights of the reactions, i.e. $S[m, r]$ is the stoichiometric weight of metabolite $m \in \mathcal{M}$ for reaction $r \in \mathcal{R}$.

Constraint-based models are inherently bipartite directed graphs and thus they can be represented graphically as Petri nets Murata (1989); Heiner et al. (2008), where places, drawn as circles, model metabolites, and transitions, drawn as squares, model reactions. The presence of an arc from a

place(transition) to a transition(place) means that the place is a reactant(product) of the reaction modelled by the transition. The weights of the arcs of the Petri net account for the stoichiometry of the constraint-based model.

**Example 2.1.** The Petri net in Figure 2b represents a constraint-based model consisting only of one reaction, the well-known reaction of equation 2a.

## 2.2 FLUX BALANCE ANALYSIS

*Flux Balance Analysis* (FBA) Orth et al. (2010) is a mathematical procedure for the estimation of steady-state fluxes in constraint-based models. FBA is generally used to predict the maximum growth rate of an organism. Let $v \in \mathbb{R}^{|\mathcal{R}|}$ be the vector of fluxes of reactions and $v[r]$ denote the flux of reaction $r$. At a steady state, it holds that $\mathcal{S} \cdot v = 0$, where $S$ is the stoichiometric matrix. Let $r_g$ be the reaction that models growth (or biomass production). Without loss of generality, it will be assumed that $L[r_g] \geq 0$. A theoretical optimum for the growth rate can be obtained by the linear programming problem (LPP) for FBA:

$$
\begin{aligned}
\max\ & v[r_g] \\
st.\ & \mathcal{S} \cdot v = 0 \\
& L \leq v \leq U
\end{aligned}
\tag{1}
$$

where the maximum $v[r_g]$ obtained by the above LPP (1) is the maximum growth rate.

## 2.3 ESSENTIAL REACTIONS

A reaction is said to be essential if it is required by the organism to grow. In other words, the deletion of an essential reaction implies null growth. Consequently, these reactions have the potential to cause the death of the modelled organism Oyelade et al. (2018).

**Definition 2.1.** Oyelade et al. (2018) A reaction $r \in \mathcal{R}$ is an *essential reaction* if the solution of the following LPP:

$$
\begin{aligned}
\max\ & v[r_g] \\
st.\ & \mathcal{S} \cdot v = 0 \\
& L \leq v \leq U \\
& v[r] = 0
\end{aligned}
\tag{2}
$$

is equal to 0 or the LPP is infeasible. In other words, a reaction $r \in \mathcal{R}$ is essential if the maximum growth rate that the model can achieve is 0 (i.e. $\max v[r_g] = 0$) when the reaction $r$ is removed (i.e. when $v[r] = 0$).

The set of essential reactions, can be computed straightforwardly by solving equation 2 for each $r \in \mathcal{R}$.

Essential reactions can also be interpreted as those reactions that are required for the production of biomass, and that lack any alternative pathway with similiar functionality. To exemplify how the topology of the network conditions the essentiality of reactions, here we provide three samples of non-essential reactions:

**Example 2.2.** Let us consider the constraint-based model in Figure 3a, where reaction $r_g$ is the biomass reaction. It can be seen that metabolite $m_2$ is required for the production of biomass. This metabolite can be both produced through reactions $r_1$ and $r_2$. If reaction $r_1$ is removed, then reaction $r_2$ can still produce metabolite $m_2$. Therefore, reaction $r_1$ is not essential. Similarly, reaction $r_2$ is not essential either.

**Example 2.3.** Let us now consider the constraint-based model in Figure 3b. Metabolite $m_4$ is a product of the biomass reaction. Once this metabolite is produced it requires to be evacuated from the network. This can be done through reaction $r_7$ or reactions $r_5$ and $r_6$. If reaction $r_7$ is removed, then reactions $r_5$ and $r_6$ can still evacuate metabolite $m_4$. Therefore, reaction $r_7$ is not essential. Similarly, reactions $r_5$ and $r_6$ are not essential either.

**Example 2.4.** Let us now consider the constraint-based model in Figure 3c. Here, metabolite $m_3$ can be diverted through the biomass reaction or through an alternative path (reactions $r_8$, $r_9$) that does not contribute to biomass production. If one of these two reactions is removed, then the other reactions can still produce metabolite $m_3$. Therefore, reactions $r_8$ and $r_9$ are not essential.

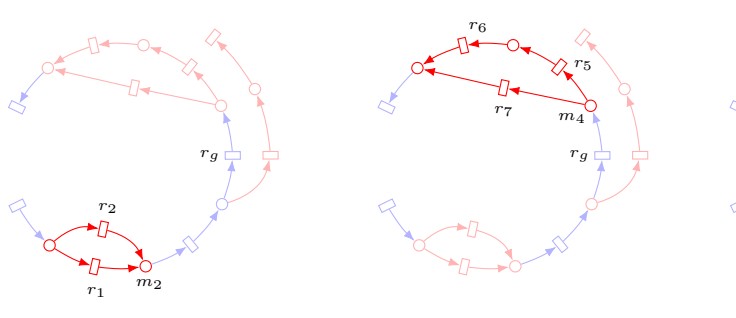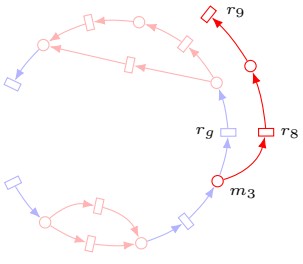

(a) Redundancy in the production of biomass.

(b) Redundancy in the evacuation of biomass products.

(c) Biomiass reachability.

Figure 3: Three sample conditions of reactions essentiality.

## 3 DIRECTED GRAPH TRANSFORMERS

In this section, we introduce all the concepts that compose a directed graph transformer model. We start by introducing the concept of graph neural networks, and directed graph neural networks and we end by introducing positional encodings for graphs. A definition of transformers models is provided in Appendix A.

### 3.1 GRAPH NEURAL NETWORKS

Graph Neural Networks (GNN) Scarselli et al. (2008); Li et al. (2015); Kipf & Welling (2016) are machine learning models that learn on data that is accompanied by a graph structure. GNNs are composed of layers of message passing networks. In each layer, the embedding vector $h_i$ of node $i$ is computed from the aggregation of the embeddings of their neighbour nodes $\mathcal{N}(i)$ of the previous layer. The initial embedding vector is usually the input feature vector that each node is given. As described in You et al. (2020), a general $k$-th GNN layer can be defined formally as:

$$h_v^{(k+1)} = \text{AGG}\left(\left\{\text{ACT}\left(\mathbf{W}^{(k)}h_u^{(k)} + b^{(k)}\right), u \in \mathcal{N}(v)\right\}\right) \tag{3}$$

where $h_v^{(k)}$ is the k-th layer embedding of node $v$, $\mathbf{W}^{(k)}$ and $b^{(k)}$ are the trainable weight matrix and bias respectively, ACT is an activation function and AGG is a commutative aggregation function such as maximisation, summation or mean. The different variations proposed to GNNs have shown to be very effective in learning on graphs data Hamilton et al. (2017); Veličković et al. (2017); Li et al. (2015). However, GNNs are known to suffer from limited expressivity Xu et al. (2018a), over-smoothing, this is, converging to a single solution after many layers Li et al. (2018), and over-squashing, this is, losing information from long-range nodes due to bottlenecks in the graph Alon & Yahav (2020). Their application to modelling long-range dependencies is thus considered limited.

### 3.2 DIRECTED GRAPH NEURAL NETWORKS

Most current GNN applications assume that graphs are undirected. However, the application of GNNs on directed graphs is not straightforward. This usually requires either transforming the directed graph into an undirected graph or propagating messages only over incoming (or outgoing) edges, which could lead to information loss. These approaches have had a great performance on benchmarks that have been historically homophilic, this is, where neighbours nodes tend to share the same label. However, the case of directed and heterophilic graphs has been less explored. To overcome this, in Rossi et al. (2023) the authors introduce the framework of Directed Graph Neural Networks (Dir-GNN). Dir-GNNs are generic GNNs that can be applied to directed graphs by aggregating messages from both the incoming and outgoing edges of each node. This extension of GNNs showed a great improvement in heterophilic datasets and seemed to outperform previous approaches for directed graphs. For this reason, we will use Dir-GNNs as the base model for our models. Formally,

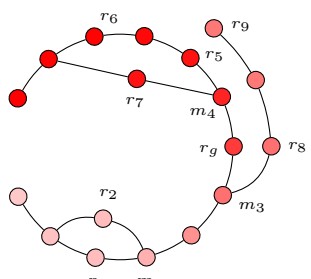 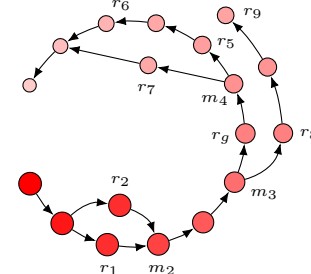

(a) Second eigenvector $v_1$ of the graph from Figure 3. Network direction has been omitted to represent that directionality is missed in the combinatorial Laplacian.

(b) First element of the eigenvectors of the magnetic Laplacian of the graph from Figure 3. The size of the nodes is proportional to the real part of the eigenvector and the color is proportional to the imaginary part.

Figure 4: Unipartite representation of the graph in Figure 3 showing the spectral information of the combinatorial and magnetic Laplacians. Here values have been normalized for visualization purposes.

the $k$-th layer can be defined as:

$$m_{u,\leftarrow}^{(k+1)} = \text{AGG}\left(\left\{\text{ACT}\left(\mathbf{W}_{\leftarrow}^{(k)} h_v^{(k)} + b_{\leftarrow}^{(k)}\right), (v,u) \in \mathcal{E}\right\}\right)$$

$$m_{u,\rightarrow}^{(k+1)} = \text{AGG}\left(\left\{\text{ACT}\left(\mathbf{W}_{\rightarrow}^{(k)} h_v^{(k)} + b_{\rightarrow}^{(k)}\right), (u,v) \in \mathcal{E}\right\}\right) \quad (4)$$

$$h_u^{(k+1)} = \text{COM}^{(k)}\left(\left\{m_{u,\leftarrow}^{(k+1)}, m_{u,\rightarrow}^{(k+1)}\right\}\right)$$

where $\mathbf{W}_{\leftarrow}^{(k)}$ and $\mathbf{W}_{\rightarrow}^{(k)}$ are the learnable weight matrices of the k-th layer, $b_{\leftarrow}^{(k)}$ and $b_{\rightarrow}^{(k)}$ are the learnable bias vectors of the k-th layer, ACT is an activation function, AGG is a commutative aggregation function, and $\text{COM}^{(k)}$ is a function that combines the incoming and outgoing messages. In this work $\text{COM}^{(k)}$ is implemented as: $\alpha^{(k)} m_{u,\leftarrow}^{(k+1)} + (1 - \alpha^{(k)}) m_{u,\rightarrow}^{(k+1)}$ with $\alpha^{(k)} \in [0,1]$ being a learnable parameter of the k-th layer.

### 3.3 POSITIONAL ENCODINGS

The attention mechanism of transformers is known to be permutation invariant. Hence, in order to capture positional information, Positional Encodings (PEs) are used. In Vaswani et al. (2017), the authors propose to add a sinusoidal PE to capture the position of each word in the sentence. However, the generalization of the sinusoidal signal to graph-structured data is not trivial. The most common approach for computing positional encodings in graph data is to use the Laplacian matrix of the graph. It is argued that the eigenvectors of the Laplacian matrix generalise the sinusoidal encodings to graphs Dwivedi & Bresson (2020).

#### 3.3.1 EIGENVECTORS OF LAPLACIAN

Let $G^{(U)} = (V,E)$, be an undirected graph without self loops, where $V$ are the nodes and $E$ the edges. The adjacency matrix $\mathbf{A}^{(U)} \in \mathbb{R}^{N \times N}$ is defined as $\mathbf{A}_{ij}^{(U)} = 1$ if $(i,j) \in E$ and $\mathbf{A}_{ij}^{(U)} = 0$ otherwise, with $N = |V|$. Given the degree of each node $d_i = \sum_{j=1}^{N} \mathbf{A}_{ij}^{(U)}$, the degree matrix is defined as $\mathbf{D} = \text{diag}(d_1, \ldots, d_N)$. The (combinatorial) Laplacian and symmetrized Laplacian are defined as:

$$\mathbf{L}_u = \mathbf{D} - \mathbf{A}^{(U)} \in \mathbb{R}^{N \times N} \quad \mathbf{L}_s = \mathbf{D}^{-1/2} \mathbf{L} \mathbf{D}^{-1/2} \in \mathbb{R}^{N \times N} \quad (5)$$

Since, the $\mathbf{L}_u$ of an undirected graph is real symmetric, $\mathbf{L}_u$ can be decomposed as $\mathbf{L}_u = \mathbf{U} \mathbf{\Lambda} \mathbf{U}^T$, where $\mathbf{U}$ is the orthonormal matrix of eigenvectors $\mathbf{U} = v_0, \ldots, v_{N-1}$ and $\mathbf{\Lambda}$ is the diagonal matrix of eigenvalues $\mathbf{\Lambda} = \lambda_0, \ldots, \lambda_{N-1}$. Given the ordered eigenvalues $\lambda_0 \leq \cdots \leq \lambda_{N-1}$, and the corresponding eigenvectors $v_0, \ldots, v_{N-1}$, it always holds that $\lambda_0 = 0$ and $v_0 = \mathbf{1}$.

In Beaini et al. (2021), it was shown that using only the eigenvector $v_1$ is enough to distinguish graphs not distinguishable by the 1-WL test. Additionally, in Kreuzer et al. (2021), the authors suggest that transformers with the full set of eigenvectors are universal approximators of the graph isomorphism problem.

**Example 3.1.** Let us consider the constraint-based model of Figure 3. The graph in Figure 4a shows the equivalent unipartite graph where the direction of the network has been omitted. In this graph, the intensity of the color of the nodes is proportional to the value of the second eigenvector of the combinatorial Laplacian $v_1$. Generally, it is known that the second eigenvector is able to represent closeness in the network, this is, nodes that are close to each other in the network will have similar values and distant nodes will have a larger difference in their values. In this sense, the combinatorial Laplacian is able to provide information of the global positioning of a node in a network. Recall that directionality is missed in the combinatorial Laplacian. However, the second eigenvector is able to represent some redundancies of the network. For instance, in this graph, it happens that nodes $r_6$ and $r_7$, which belong to a redundant path, have similar values in the second eigenvector (i.e. $v_{1,r6} \approx v_{1,r7}$). The same case happens with nodes $r_1$ and $r_2$ (i.e. $v_{1,r1} \approx v_{1,r2}$).

### 3.3.2 MAGNETIC LAPLACIAN

As it was pointed out in Furutani et al. (2020) and Geisler et al. (2023), the combinatorial Laplacian fails to distinguish directionality on graphs. To address this issue, the authors propose the use of the eigenvectors of the magnetic Laplacian, where in addition to the connectivity of the network, the directionality of the network is encoded in the complex plane. Let $G^{(D)} = (V, E)$ be a directed graph with adjacency matrix $\mathbf{A}^{(D)}$, where $\mathbf{A}_{ij}^{(D)} = 1$ if $(i, j) \in E$ and $\mathbf{A}_{ij}^{(D)} = 0$ otherwise. Additionally, let $\mathbf{A}^{(U)}$ be the adjacency matrix of the undirected version of $G^{(D)}$. The magnetic Laplacian is defined as:

$$\mathbf{L}_m = \mathbf{D} - \mathbf{\Gamma_m} \odot \mathbf{A}^{(U)} \in \mathbb{C}^{N \times N} \tag{6}$$

where $\odot$ is the Hadamard product and $\mathbf{\Gamma_m} \in \mathbb{C}^{N \times N}$ is a Hermitian matrix whose $(i, j)$ is equal to: $e^{i2\pi q(\mathbf{A}_{ij}^{(D)} - \mathbf{A}_{ji}^{(D)})}$, where $q \in [0, 1)$ is a rotation parameter or potential. Notice that when $q = 0$, the magnetic Laplacian is equal to the combinatorial Laplacian. In addition, notice that $e^{i2\pi q(\mathbf{A}_{ij}^{(D)} - \mathbf{A}_{ji}^{(D)})}$ encodes the direction of the edge $(i, j)$. For undirected edges, i.e. $\mathbf{A}_{ij}^{(D)} = \mathbf{A}_{ji}^{(D)} = 1$, the value is equal to 1. For directed edges, i.e. $\mathbf{A}_{ij}^{(D)} = 1, \mathbf{A}_{ji}^{(D)} = 0$ and $\mathbf{A}_{ij}^{(D)} = 0, \mathbf{A}_{ji}^{(D)} = 1$, the value is equal to $e^{i2\pi q}$ and $e^{-i2\pi q}$ respectively. From here onwards, we will assume the eigenvalues of the magnetic Laplacian to be ordered: $0 = \lambda_0 \leq \lambda_1 \leq \cdots \leq \lambda_{N-1}$.

**Example 3.2.** Let us consider again the constraint-based model of Figure 3. The graph in Figure 4b shows the equivalent unipartite graph. In this graph, the size of the nodes is proportional to the real part of the first eigenvector of the magnetic Laplacian, denoted $\text{Re}(\phi_0^{(q)})$, and the color of the nodes is proportional to the imaginary part $\text{Im}(\phi_0^{(q)})$. As it is shown in Furutani et al. (2020); Geisler et al. (2023), and as it can be seen in the graph, the magnetic Laplacian provides a topological sorting of the network in the complex plane. This is particularly useful in the case of metabolic networks, as it is able to capture redundancies in the network while also considering the directionality of the network.

For instance, notice that, in this graph, nodes $r_1$ and $r_2$, which are both part of a redundant path, have the same color and size (i.e. $\text{Re}(\phi_{0,r_1}^{(q)}) \approx \text{Re}(\phi_{0,r_2}^{(q)})$ and $\text{Im}(\phi_{0,r_1}^{(q)}) \approx \text{Im}(\phi_{0,r_2}^{(q)})$). The same case happens with nodes $r_3$, $r_4$ and $r_5$. Notice also that, unlike the combinatorial Laplacian, the magnetic Laplacian is able to capture the fact that paths $m_3 \to r_g$ and $m_3 \to r_8$ are parallel.

### 3.3.3 NOTATION

To denote the set of eigenvalues and eigenvectors of the magnetic Laplacian, we will use the following notation: $\mathbf{\Lambda}^{(q)} = \lambda_0^{(q)}, \ldots, \lambda_{N-1}^{(q)}$ and $\mathbf{\Phi}^{(q)} = \phi_0^{(q)}, \ldots, \phi_{N-1}^{(q)}$, where $\phi_i^{(q)}$ is the $i$-th eigenvector of the magnetic Laplacian for rotation parameter $q$. We use $\phi_{i,j}^{(q)}$ to denote the entry of the $i$-th eigenvector corresponding with the $j$-th node.

We will use the fact that the magnetic Laplacian with $q = 0$ is equivalent to the combinatorial Laplacian to denote the eigenvalues and eigenvectors of the combinatorial Laplacian as $\mathbf{\Lambda}^{(0)} =$

$\lambda_0^{(0)}, \ldots, \lambda_{N-1}^{(0)}$ and $\mathbf{\Phi}^{(0)} = \phi_0^{(0)}, \ldots, \phi_{N-1}^{(0)}$ respectively. Similarly, we denote $\phi_{i,j}^{(0)}$ as the entry of the $i$-th eigenvector of the combinatorial Laplacian corresponding with the $j$-th node.

## 4 ARCHITECTURE

In this section, we present the organization of the proposed metabolic transformer. The architecture is a directed graph transformer, which is composed of a positional encoding layer (that may include directional information) and a graph transformer layer with directed message passing.

This architecture is motivated by the fact that message passing alone is not enough to identify the sufficient conditions that were introduced in Section 3. In Appendix B.1 an example is provided where the asymptotic complexity of identifying redundancies in a network is of $O(D)$ message passing steps, with $D$ being the diameter of the network. Another significant property of this architecture is that it is able to capture directionality in the network. Again, it is easy to see that identifying redundancies in a network requires directional information. An example of this is provided in Appendix B.2.

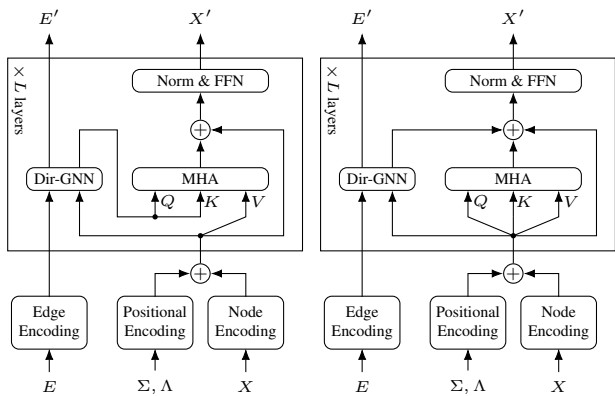

(a) Directed graph transformer based on the SAT architecture Chen et al. (2022). (b) Directed graph transformer based on the GPS architecture Rampášek et al. (2022).

Figure 5: Two proposed architectures for the directed graph transformer.

### 4.1 SPECTRAL ENCODING

As mentioned before, bipartite metabolic networks will be treated as non-bipartite networks. This will allow us to compute spectral positional encodings as described before. This procedure is detailed in Appendix G. In Section 3 we described how the eigenvalues and eigenvectors of the combinatorial Laplacian and the magnetic Laplacian capture information about the global position of nodes in a graph. To be able to learn useful representations from this information, we will use the Laplacian Positional Encoder (LapPE), introduced in Kreuzer et al. (2021) and the Magnetic Laplacian Positional Encoder (MagLapPE), introduced in Geisler et al. (2023). A detailed description of these positional encoders is provided in Appendix C.

### 4.2 METABOLIC TRANSFORMER

Now that the positional encoding layers are introduced, we are ready to introduce the metabolic transformer layer. The approach taken is a combination of a directed message passing layer to capture the local structure of the network, and a self-attention layer over the whole set of nodes to capture global relations in the network. This combination of mesage passing with global attention has been presented before Chen et al. (2022); Rampášek et al. (2022), with promising results in many graph-related tasks. These works however, were limited to undirected graphs. In this work, two different flavours of graph transformers have been extended to include directed message passing.

1. The first approach (SAT), follows the work of Chen et al. (2022) and can be seen in Figure 5a. This approach is intended to capture structural similarities through the attention mechanism. The layer achieves this by using the GNN output as the query and the key of the attention mechanism, and the graph embeddings as the value. The output of the attention mechanism is used then as the input of the next layer. Formally, the $l-$th layer of the graph transformer

is defined as follows:

$$\mathbf{X}_G^{(l+1)} = \text{Dir-GNN}(\mathbf{X}^{(l)}, \mathbf{E}^{(l)}, \mathbf{A}^{(D)})$$
$$\mathbf{X}_A^{(l+1)} = \text{MHA}(\mathbf{X}_G^{(l+1)}, \mathbf{X}_G^{(l+1)}, X^{(l+1)}) \qquad (7)$$
$$\mathbf{X}^{(l+1)} = \text{Norm}(\text{FFN}(\mathbf{X}_A^{(l+1)} + \mathbf{X}^{(l)}))$$

where $\mathbf{X}^{(l+1)} \in \mathbb{R}^{N \times d}$ is the output of the $l-$th layer, $\mathbf{A}^{(D)}$ is the adjacency matrix of the graph, $\mathbf{E}^{(0)}$ are the initial edge features, and $\mathbf{X}^{(0)}$ are the sum of the initial graph features and positional encoder features.

2. The second approach (GPS), presented in Rampášek et al. (2022), is depicted in Figure 5b. This approach applies simultaneously a GNN layer, that captures the local information of the network, and a self-attention layer over the whole set of nodes embeddings. The output of both layers is added and the output of the layer is passed as input to the next layer. Formally, the $l-$th layer of the graph transformer is defined as follows:

$$\mathbf{X}_G^{(l+1)} = \text{Dir-GNN}(\mathbf{X}^{(l)}, \mathbf{E}^{(l)}, \mathbf{A}^{(D)})$$
$$\mathbf{X}_A^{(l+1)} = \text{MHA}(\mathbf{X}^{(l)}) \qquad (8)$$
$$\mathbf{X}^{(l+1)} = \text{Norm}(\text{FFN}(\mathbf{X}_G^{(l+1)} + \mathbf{X}_A^{(l+1)} + \mathbf{X}^{(l)}))$$

where $\mathbf{X}^{(l+1)} \in \mathbb{R}^{N \times d}$ is the output of the $l-$th layer, $\mathbf{A}^{(D)}$ is the adjacency matrix of the graph, $\mathbf{E}^{(0)}$ are the initial edge features, and $\mathbf{X}^{(0)}$ are the sum of the initial graph features and positional encoder features.

## 5 RESULTS

This section reports the results obtained in a dataset of 23k genome-scale models with an average of 2k node. Since no previous dataset exists for GEM, we are releasing the curated dataset along with this publication. The detailes of the dataset curation can be found in Section E. The results presented here include those obtained with the proposed graph transformer models with the different positional encoders. We are particularly interested in studying the effect of the different positional encoders and different GNN layers on the overall performance. To do so, we compare the results obtained with baseline GNN models, which include: Graph Convolutional Networks (GCN) Kipf &

| Model | Params | Epoch time (s) | F1 ↑ |
|---|---|---|---|
| SAT+LapPE+Dir-GCN | 298393 | 619.71 | 0.6622 |
| SAT+LapPE+Dir-GAT | 349849 | 809.96 | 0.6328 |
| SAT+LapPE+Dir-GINE | 398209 | 757.61 | 0.5651 |
| SAT+LapPE+Dir-Gated | 498049 | 824.91 | 0.6628 |
| SAT+MagLapPE+Dir-GCN | 269145 | 614.65 | 0.6370 |
| SAT+MagLapPE+Dir-GAT | 320601 | 813.43 | 0.6786 |
| SAT+MagLapPE+Dir-GINE | 368961 | 735.50 | 0.5877 |
| SAT+MagLapPE+Dir-Gated | 468801 | 639.33 | 0.6974 |
| GPS+LapPE+Dir-GCN | 298393 | 557.39 | 0.6659 |
| GPS+LapPE+Dir-GAT | 349849 | 646.62 | 0.6718 |
| GPS+LapPE+Dir-GINE | 398209 | 587.89 | 0.6195 |
| GPS+LapPE+Dir-Gated | 498049 | 940.53 | **0.7145** |
| GPS+MagLapPE+Dir-GCN | 269145 | 546.68 | 0.6589 |
| GPS+MagLapPE+Dir-GAT | 320601 | 712.57 | 0.6708 |
| GPS+MagLapPE+Dir-GINE | 368961 | 735.50 | 0.5984 |
| GPS+MagLapPE+Dir-Gated | 468801 | 657.18 | 0.6841 |

Table 1: Results of the different models on the test set.

Welling (2016), Graph Attention Networks (GAT) Veličković et al. (2017), Graph Isomorphism Networks (GINE) Xu et al. (2018a), and Gated Graph Neural Networks (Gated) Li et al. (2015). For each of the baseline models, we used the directed version of the model, as described in Section 4. Details on the hyperparameters and experimental setup can be found in Appendix F.

| Model | F1 (+) ↑ | ROC AUC (+) ↑ | F1 (-) ↑ | ROC AUC (-) ↑ |
|---|---|---|---|---|
| FlowGAT Hasibi et al. (2024) | $\mathbf{0.85 \pm 0.033}$ | $0.495 \pm 0.053$ | $0.02 \pm 0.04$ | $0.525 \pm 0.075$ |
| Metabolic Transformer (ours) | $0.845 \pm 0.027$ | $\mathbf{0.857 \pm 0.022}$ | $\mathbf{0.69 \pm 0.026}$ | $\mathbf{0.849 \pm 0.017}$ |

Table 2: Performance comparison in synthetic reaction essentiality prediction on *E.coli* model iML1515 Monk et al. (2017). Here (+) indicates essentiality as positive class, while (-) indicates the opposite problem, this is, prediction of non-essentiality.

| Model | F1 (+) ↑ | ROC AUC (+) ↑ | F1 (-) ↑ | ROC AUC (-) ↑ |
|---|---|---|---|---|
| FlowGAT Hasibi et al. (2024) | **0.86 ± 0.015** | 0.56 ± 0.121 | 0.161 ± 0.099 | 0.587 ± 0.134 |
| Metabolic Transformer (ours) | 0.68 ± 0.031 | **0.845 ± 0.03** | **0.624 ± 0.093** | **0.797 ± 0.083** |

Table 3: Performance comparison in real gene essentiality prediction on *E.coli* model iML1515 Monk et al. (2017). Here (+) indicates essentiality as positive class, while (-) indicates the opposite problem, this is, prediction of non-essentiality.

As it is explained in Appendix E, the generated dataset is highly imbalanced, with an average of 8.8% of positive samples. To account for this, we report the F1 score of the models to compare their performance. From the results shown in Table 1, we can see that there is a clear dependence of the results on the message passing scheme used. Overall, the results suggest that message passing layers that aggregate the messages in a non-linear way (Dir-GAT, Dir-Gated) perform better than those that weight each message contribution the same (Dir-GCN, Dir-GINE). This suggests that the solution suffers from the well-known over-squashing problem Di Giovanni et al. (2023). Regarding the graph transformer architecture, generally, GPS-based solutions seem to perform slightly better than SAT-based solutions. If we compare now the results obtained with the different positional encoders, surprisingly, there is no clear advantage of using one over the other. It seems that the MagLapPE performs better with the SAT-based solutions, while the LapPE performs better with the GPS-based solutions. The best results are obtained with the GPS + LapPE + Dir-Gated model. Recall that LapPE does not capture the directional information of the graph. Despite this, LapPE is equally competitive than MagLapPE and outperforms MagLapPE in the case mentioned. This might suggest that (i) direction information is not as important for the task at hand or (ii) the information captured with the message passing is enough to capture the directional information of the graph. These results also suggest that, for this task, the use of the message passing scheme seems to be more important than the positional encoding scheme.

In Appendix D we provide an ablation study of the different components of the proposed architecture. A performance comparison between GPU and CPU inference for essential reactions computation is presented in Appendix H

In order to evaluate the performance of the metabolic transformer against the existing state-of-the-art, we compared the results obtained with the FlowGAT model Hasibi et al. (2024). As metabolic transformer, here we used the best-performing model according to Table 1, this is, GPS+LapPE+Dir-Gated. Table 2 show the results obtained in synthetic knock-out essential reaction prediction, using the iML1515 model of *Escherichia coli* Monk et al. (2017). As mentioned previously in this work, since FlowGAT relies in locality information, the capability to model essential reactions is limited, which explains the difference in performance. Table 3 shows the results obtained in real gene essentiality prediction using the same model. A fine-tuning was performed with the metabolic transformer to adapt the model to the task at hand. The results show that FlowGAT shows a higher F1 score, while performing poorly in the ROC AUC metric. In order to clarify this divergence in scores, confusion matrices from both problems are shown in Figures 7 and 6. In this Figure it can be seen that FlowGAT tends to classify all reactions as positive, while almost no discrimination is done with negative classes. Since the

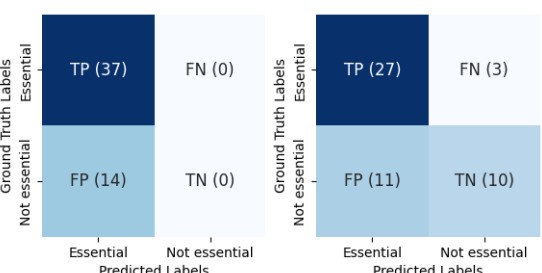

(a) FlowGAT Hasibi et al. (2024)   (b) Metabolic Transformer (ours)

Figure 6: Confusion matrix for synthetic reaction essentiality prediction on *E.coli* model iML1515 Monk et al. (2017), obtained from best F1-score.

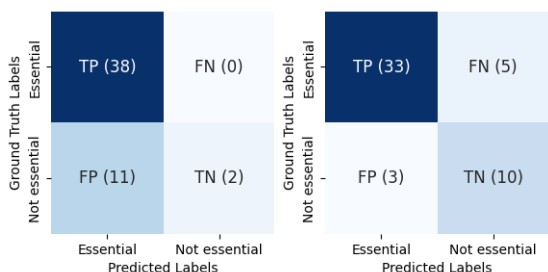

(a) FlowGAT Hasibi et al. (2024)   (b) Metabolic Transformer (ours)

Figure 7: Confusion matrix for real gene essentiality prediction on *E.coli* model iML1515 Monk et al. (2017), obtained from best F1-score.

postive class here is the majority class, the F1 score obtained is large, while the ROC AUC shows that discriminative power of FlowGAT is considerably lower than the metabolic transformer introduced in this work. We note that the evaluation dataset used consists of a small number of samples with a high presence of positive samples, which explains the fluctuation in the ROC AUC metric. Note also that, while FlowGAT is a specialized model narrowed down for this particular tasks, our approach is a general model finetuned for gene essentiality prediction.

## 6 CONCLUSIONS

This work proposes the metabolic trasnformers, a novel approach for the prediction of essential reactions in genome-scale models by using directed graph transformers, entirely trained with synthetic knock-out data. The proposed approach is based on the use of directed GNNs as the base model to capture local information, and the use of positional encodings based on the combinatorial Laplacian and the magnetic Laplacian. To compare the performance of the proposed approach, we built a dataset of genome-scale models from the metabolism of different microorganisms from public databases. Aditionally, we compared the performance of the proposed approach with existing methods for essential reactions and genes prediction, and we showed that the proposed approach achieves state-of-the-art performance in both tasks. The study on the architectural choices show that the performance is highly dependent on the message passing scheme used. In particular, the best results are obtained with the use of gated GNNs with a GPS architecture and Laplacian-based positional encodings. The results also show no clear advantage of using magnetic Laplacian-based positional encodings over Laplacian-based positional encodings, when combined with directed message passing. This work, however, is fundamentally limited by the scalability of the model, both from a data and memory perspective. Transformers are well known to have a $O(N^2)$ memory complexity. In our case, we used graphs from 113 microorganisms with an average of 2190 nodes, with the largest one having 5861 nodes. Currently, the largest available genome-scale model is the human model Recon3D Brunk et al. (2018) with 17683 nodes, which poses a challenge to the scalability of the model. Possible solutions to this issue include using more efficient attention mechanisms Choromanski et al. (2020)

## 7 ETHICS STATEMENT

The contribution of this work is to provide a novel approach to identify essential reactions in GEMs. This has the potential to facilitate the identification of essential reactions in a broad range of automatically reconstructed GEMs, a task previously deemed infeasible or requiring extensive manual curation of models. Current approaches that identify essential reactions in GEMs are aimed at contributing to the development of new antibiotics or cancer treatments, among others. Therefore we do not expect any ethical or societal issues to arise from this work.

## 8 REPRODUCIBILITY STATEMENT

Both the code and datasets requried for the reproducibility of the results reported will be made available under GNU-GPLv3 license upon acceptance of the paper. Aditionally, the steps taken to generate the dataset and hyperparameters used can be found in the Appendix F, E and G.

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

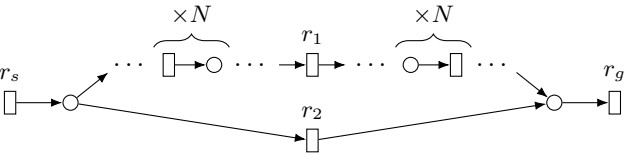

Figure 8: Example Petri net where $r_1$ and $r_2$ belong to redundant pahts. A total of $2N+2$ message passing steps are required to reach $r_2$ from $r_1$.

## A    TRANSFORMERS

Transformers are neural network models that have been shown to be very effective in a wide range of tasks, ranging from language modelling Vaswani et al. (2017), to computer vision Dosovitskiy et al. (2020) or approximating combinatorial problems such as the Travelling Salesman Problem Kool et al. (2018). Transformers are composed of two main blocks: self-attention and feed-forward layers. The self-attention layer is a function that takes as input the $N$ node features $\mathbf{X} \in \mathbb{R}^{N \times d}$, it linearly projects $\mathbf{X}$ into the query $\mathbf{Q}$, key $\mathbf{K}$ and value $\mathbf{V}$ this is, $\mathbf{Q} = \mathbf{X}\mathbf{W}_Q$, $\mathbf{K} = \mathbf{X}\mathbf{W}_K$ and $\mathbf{V} = \mathbf{X}\mathbf{W}_V$, and computes self-attention as:

$$\text{SelfAttention}(\mathbf{Q}, \mathbf{K}, \mathbf{V}) = \text{softmax}\left(\frac{\mathbf{Q}\mathbf{K}^T}{\sqrt{d}}\right)\mathbf{V} \in \mathbb{R}^{N \times d} \tag{9}$$

where $d$ is the dimension of $\mathbf{Q}$ and $\mathbf{W}_Q, \mathbf{W}_K, \mathbf{W}_V \in \mathbb{R}^{d \times d}$ are trainable weight matrices. Generally, multi-head attention is used, this is, the self-attention is computed $h$ times with different weight matrices, and the resulting outputs are concatenated. The multi-head attention layer is then followed by a feed-forward layer. The overall transformer architecture can be defined as:

$$\mathbf{X}' = \mathbf{X} + \text{MHA}(\mathbf{X})$$
$$\text{Transformer}(\mathbf{X}) = \text{FFN}(\mathbf{X}') = \text{ReLU}(\mathbf{X}'\mathbf{W}_1)\mathbf{W}_2 \tag{10}$$

where MHA is the multi-head attention layer, ReLU is the activation function, and $\mathbf{W}_1, \mathbf{W}_2 \in \mathbb{R}^{d \times d}$ are trainable weight matrices. In the above equation, $\text{MHA}(\mathbf{X}, \mathbf{X}, \mathbf{X})$ has been written as $\text{MHA}(\mathbf{X})$ for simplicity.

## B    CHALLENGES

This section provides an insight into the existing challenges to identify redundancies in metabolic networks, and how these challenges have driven our architectural decisions.

### B.1    LONG-RANGE DEPENDENT

It is straightforward to check that the message passing framework is constrained when it comes to identifying redundancies on networks. To see this, consider the example network of Figure 8. In this network, reaction $r_s$ is a source reaction in the network, and reaction $r_g$ is the objective reaction. It is clear that there are two different paths from $r_s$ to $r_g$, one path that goes through reaction $r_1$ and another path that goes through reaction $r_2$. This makes reactions $r_1$ and $r_2$ not essential. Notice that between reaction $r_1$ and $r_2$ there are a total of $2N + 2$ nodes. This means that, if we wanted to transfer information from $r_1$ to $r_2$ using message passing, we would need at least $2N + 2$ steps (assuming undirected message passing). Ultimately, this means that, for reaction $r_1$ to *acknowledge* the existence of reaction $r_2$, it would need to pass information through at least $2N + 2$ steps. Therefore, it is clear that the number of steps required to identify redundancies in the network scales linearly with the number of nodes in the network, this is, given a network $N$, the worst case number of steps required is $O(D)$, where $D$ is the diameter of $N$.

Given this limitation of message passing, we need to look for a different mechanism for capturing the global information of the network. This shortcoming in message passing may be addressed by using positional encodings Dwivedi & Bresson (2020). Since positional encodings try to capture the global position of a node in the network, they may reduce the number of steps required to identify redundancies in the network. However, as shown in Section D, GNNs with positional encodings are not powerful enough to identify redundancies in the network.

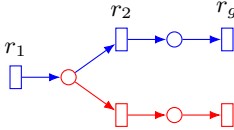 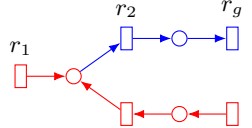 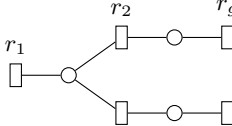

(a) Example Petri net where reactions $r_1$ and $r_2$ are essential reactions.

(b) Example Petri net where reaction $r_1$ is not essential and reaction $r_2$ is essential.

(c) Undirected graph whose topology matches the one of Figures (a) and (b).

Figure 9: The directionality of the network determines the essentiality of the reactions. Both networks in Figures (a) and (b) share the same undirected topology, but the directionality of the edges is different. This makes reaction $r_1$ essential in Figure (a) and not essential in Figure (b).

### B.2 DIRECTIONALITY

Given the wide application of graph learning methods to undirected graphs, one might be tempted to use undirected graphs to represent metabolic networks. Again, it is straightforward to show that the directionality of the network is a key property in the identification of redundancies and that ignoring it leads to a loss of information in the network. To see this, consider the Petri nets of Figures 9a and 9b, where reaction $r_g$ is the objective reaction. Notice that, given the direction of the reactions, reaction $r_1$ is essential in Figure 9a but not essential in Figure 9b, since in the last, alternative reactions exist that are able to supply the objective reaction. Notice that, if we ignore the directionality of the reactions, both networks share the same topology, depicted in Figure 9c.

In this graph, reaction $r_2$ is always involved in a path that leads to the objective reaction $r_g$. Therefore, it is easy to infer that this reaction will always be essential, even if we ignore the directionality of the reactions. However, without directionality, it is impossible to infer whether reaction $r_1$ is essential or not. It is clear then, that any approach that dismisses the direction of the network will suffer a loss of key information.

## C SPECTRAL ENCODING

This section provides a detailed description of the positional encoding proposed in Section 4. In particular, we will definitions are provided for the combinatorial Laplacian Positional Encoder (LapPE) Kreuzer et al. (2021) and the Magnetic Laplacian Positional Encoder (MagLapPE) Geisler et al. (2023).

### C.1 LAPLACIAN POSITIONAL ENCODER

To encode the spectral information of the combinatorial Laplacian we use an approach similar to the one in Kreuzer et al. (2021). In this work, the first $k$ eigenvectors with the minimum eigenvalues are selected. These eigenvectors are concatenated with their corresponding eigenvalues, passed through a linear layer, a self-attention layer and finally each embedding is pooled node-wise through a sum pooling. More formally, given the $k$ eigenvectors corresponding to the node $i$, which we denote as $\phi_{:k,i}$, and the $k$ eigenvalues, which we denote as $\lambda_{:k}$, the Laplacian Positional encoding for node $i$ is defined as follows:

$$
X_i = \text{FFN}\left(\begin{bmatrix} \phi_{:k,i} \\ \lambda_{:k}^T \end{bmatrix}\right)
$$
$$
X_i' = \text{MHA}(X_i)
$$
$$
X_i'' = \text{FFN}\left(\sum_j^k X_{i,j}'\right)
$$
(11)

where $X_i'' \in \mathbb{R}^d$ is the final embedding of the node $i$, with $d$ being the dimension of the embedding. Normalization has been omitted from the above definition. To handle the sign invariance of the eigenvectors, the sign of $\phi_{:k,i}$ is randomly flipped, as it was done in Dwivedi & Bresson (2020).

## C.2 MAGNETIC LAPLACIAN POSITIONAL ENCODER

To encode the spectral information of the magnetic Laplacian, we will use the positional encoder proposed in Geisler et al. (2023). Similarly as in the Laplacian Positional Encoder, the first $k$ eigenvectors with the minimum eigenvalues are selected. Let us denote, $\phi_{:k,i}$ as the $k$ eigenvectors corresponding to the node $i$, and $\lambda_{:k}$ as the $k$ eigenvalues. The Magnetic Laplacian Positional Encoder is defined as follows:

$$
\begin{aligned}
X_i &= \text{FFN}\left(\text{Re}(\phi_{:k,i}) \parallel \text{Im}(\phi_{:k,i})\right) \parallel \lambda_{:k}^T \\
X_i' &= \text{MHA}(X_i) + X_i \\
\mathbf{X}'' &= \text{FFN}\left(\overset{N}{\underset{i=1}{\parallel}} X_i'\right) \\
\mathbf{X}''' &= \text{MHA}(\mathbf{X}'') + \mathbf{X}''
\end{aligned}
\tag{12}
$$

where $\mathbf{X}''' \in \mathbb{R}^{N \times d}$ is the final matrix of embeddings of the $N$ nodes, with $d$ being the dimension of the embedding. Normalization and dropout have been omitted from the definition. Notice that, unlike the Laplacian Positional Encoder, this encoder includes skip connections after the self-attention layer, and performs self-attention before and after concatenating the embeddings of the nodes. Empirically, we did not observe any significant advantage of using self-attention. To handle the sign invariance of the eigenvectors, we use the same approach proposed in Geisler et al. (2023), this is, the sign of each eigenvector is determined such that the maximum real magnitude is positive. The authors also propose the use of SignNet Lim et al. (2022) to handle the sign invariance, however, empirically we did not observe any significant advantage using SignNet. The eigenvector normalization is performed as described in the original paper Geisler et al. (2023).

# D ABLATION STUDY

In this section, we perform an ablation study to show the importance of the different components of the proposed model. In particular, we will show the performance obtained without the use of message passing, without the use of transformers, and the use of directed GNNs.

| Model | Params | Epoch time (s) | F1 $\uparrow$ |
|---|---|---|---|
| Transformer + MagLapPE | 482305 | 548.12 | 0.4166 |

Table 4: Ablation without message passing.

First, let us consider the model without the use of message passing, this is, using only a transformer layer and the positional encodings. Here we used directly magnetic Laplacian positional encodings, since combinatorial Laplacian positional encodings are not able to capture directionality. The results are shown in Table 4. Here we can see that the model performance is not as good, with an F1 score of 0.4166. Based on this, it seems that the model struggles to capture local information relying only on the positional encodings.

| Model | Params | Epoch time (s) | F1 $\uparrow$ |
|---|---|---|---|
| SAT + LapPE + Gated | 374785 | 642.65 | 0.6003 |
| GPS + LapPE + Gated | 374785 | 581.06 | 0.6006 |
| SAT + MagLapPE + Gated | 345537 | 607.56 | 0.6022 |
| GPS + MagLapPE + Gated | 345537 | 553.92 | 0.5524 |

Table 5: Ablation using non-directed message passing.

Let us now consider the model without directed GNNs. Here we used both the SAT and the GPS architecture with an undirected gated GNN layer. We resort to the gated GNN layer since it shows a better performance than other GNN layers in our given dataset. The results are shown in Table 5. Here we can see that the model shows a better performance, reaching up to 0.6022 F1 score. Notice

that this outperforms some of the models presented in Table 1. However, it is still far from the best performance obtained on the task. Compared to the results in Table 4, there is a clear improvement in the score. This suggests that a great performance can be obtained in the task even without the directionality information.

| Model | Params | Epoch time (s) | F1 $\uparrow$ |
|---|---|---|---|
| LapPE + Dir-Gated | 662785 | 167.62 | 0.6445 |
| MagLapPE + Dir-Gated | 591169 | 166.88 | 0.6512 |

Table 6: Ablation without transformer.

Finally, let us consider the model without the use of transformers. Here we used directly a sequence of GNN layers without any transformer layer. For this task, we used a directed gated GNN layer. The results are shown in Table 6. Here we can see that the model shows a surprisingly good performance, reaching up to 0.6512 F1 score. This performance again outperforms some of the models presented in Table 1, however, it is still far from the best performance obtained on the task. This highlights the importance of the message passing layer and, in particular, the importance of the directed GNN layer. Finally, it shows that, without the global information provided by the transformer, the model is still not able to fully capture the redundancies in the network.

## E  DATASET

To generate the dataset used in this work, we gathered 197 genome-scale models from the BiGG King et al. (2016) and Biomodels Malik-Sheriff et al. (2020) databases. The models correspond with a total of 113 different organisms, having multiple strains for some of them. The amount of metabolites of the models ranges from 7 to 2038 metabolites with an average of 772. The amount of reactions ranges from 6 to 4047 reactions with an average of 1115. The largest model contains a total of 5861 nodes. The number of essential reactions is on average 26% of the total reactions with a standard deviation of 0.17. If we compare the number of essential reactions with the total number of nodes, on average, 14% of the nodes are essential reactions with a standard deviation of 0.14. It can be seen that the amount of essential reactions is highly imbalanced, with an average of 218 essential reactions. To generate our dataset the models were split into 153 models for training, 21 models for validation and 21 models for testing. Since models of the same organism tend to share more similar network topologies, models of the same organism were not split into different sets. The split between training, validation and test sets was done manually to ensure that the models in all sets have a balanced number of reactions, metabolites, and essential reactions, as well as the maximum degree of the nodes and the degree of the objective reaction. The detailed list of models and splits is available as supplementary material.

**Essential reactions computation** To compute the essential reactions of each model, we used the COBRApy Ebrahim et al. (2013) framework. For all the models, we remove restrictions that enforce a non-null flux in the reactions (i.e. we set $L = 0$), this is, we assume a rich medium and we impose no constraint on growth production or any other metabolite. Notice that, the proposed method in this work does not limit the possibility of adding such restrictions for reactions. Future approaches could include reaction fluxes as additional features.

**Data augmentation** Since we are able to compute essential reactions of any model, we performed random modifications in the models as a way to augment the dataset. To generate a modified model, we performed randomly two types of modifications: (i) we randomly chose any reaction of the model and used it as the objective reaction and (ii) we took the biomass reaction and randomly added or removed between 1 and 10 reactants and between 1 and 10 products of the model. In the case of producing an infeasible problem, the generated model was discarded. Notice that both presented modifications alter the objective function and therefore the computed essential reactions. The resulting dataset after the data augmentation process contained a total of 18703 samples with 14419 train samples, 2142 validation samples and 2142 test samples, with 2190 nodes on average and an 8.8% of essential reactions on average. This dataset is particularly interesting as it contains a large number of graphs, which is a feature more common in graph-prediction tasks, but on average, it

also contains a large number of nodes for node-prediction. In addition, the node label depends on the global structure of the graph.

## F  EXPERIMENTAL SETUP

In this section, we describe the experimental setup used to train and evaluate the models. In addition to the architectural decision described in Section 4, we also used jumping knowledge Xu et al. (2018b) with *max* aggregation and skip connections with GNNs (ommited in Figure 5). In the case of GAT, we used 4 attention heads. Edge features were included in all GNN models except for GCN. All the models were trained with the AdamW optimizer Loshchilov & Hutter (2017).

The training was performed using NVIDIA A10 and NVIDIA Geforce RTX 3090 GPUs, both with 24 GB RAM. Given the size of the data samples, the maximum batch size that could accommodate the GPU memory was 3 samples. The number of epochs was set to a limit of 100 epochs with 10 warmup epochs. The training of a single graph transformer model took around 24 hours.

**Hyperparameters**  Given the computational cost of training the models, we performed a limited hyperparameter search. Generally, we optimized each of the hyperparameters in a greedy fashion, this is, we fixed the rest of the hyperparameters and optimized one at a time. Table 7 shows the main hyperparameters used for training the models.

| Model | Hyperparameter | Value |
|---|---|---|
| Base | Layers | 6 |
| | FNN layers | 2 |
| | Hidden dim | 64 |
| | Heads | 4 |
| | Dropout | 0.0 |
| | GNN Aggr | Sum |
| | Norm | Layer Norm |
| Optimizer | Learning rate | 0.0008 |
| | Weight decay | $1 \times 10^{-5}$ |
| Laplacian Encoder | Freq. | 10 minimum eigv. |
| | Use attention | Yes (layers = 1, heads = 2) |
| | FFN layers | 2 |
| Magnetic Laplacian Encoder | Freq. | 10 minimum eigv. |
| | Use attention | No |
| | FFN layers | 2 |

Table 7: Hyperparameters used for graph transformer models.

For the ablation studies presented in Section D, in the case of the model without message passing, we used the hyperparameters in Table 8. In the case of graph transformers with undirected GNNs, we used the same hyperparameters as in Table 7. Finally, in the case of the model without transformers, we used the hyperparameters in Table 9. For the latter, we used 350 epochs instead of 100.

The FlowGAT model was trained using the same code and hyperparameters provided in the original paper Hasibi et al. (2024). The metabolic trasnformer was finetuned for the *E.coli* model using the above hyperparameters, learning rate of 0.0001 and freezing the transformer layers, except for the two prediction heads.

**Loss function**  Since we are training a binary classification model, we used binary cross-entropy as loss function. Through this work, we have dealt with the prediction of reaction essentiality in a bipartite graph. Clearly, the loss $\mathcal{L}(y, \hat{y})$ was computed only for reaction nodes and not for metabolites nodes. This is:

$$\mathcal{L}(y, \hat{y}) = -\sum_{i \in \mathcal{R}} y_i \log \hat{y}_i + (1 - y_i) \log(1 - \hat{y}_i) \tag{13}$$

| Model | Hyperparameter | Value |
|---|---|---|
| Base | Layers | 6 |
| | FNN layers | 2 |
| | Hidden dim | 96 |
| | Heads | 4 |
| | Dropout | 0.0 |
| | Norm | Layer Norm |
| Optimizer | Learning rate | 0.001 |
| | Weight decay | $1 \times 10^{-5}$ |
| Magnetic Laplacian Encoder | Freq. | 10 minimum eigv. |
| | Use attention | No |
| | FFN layers | 2 |

Table 8: Hyperparameters used for transformer models.

| Model | Hyperparameter | Value |
|---|---|---|
| Base | Layers | 6 |
| | Hidden dim | 96 |
| | Dropout | 0.0 |
| | GNN Aggr | Sum |
| | Norm | Layer Norm |
| Optimizer | Learning rate | 0.0008 |
| | Weight decay | 0.0 |
| Laplacian Encoder | Freq. | 10 minimum eigv. |
| | Use attention | Yes (layers = 1, heads = 2) |
| | FFN layers | 2 |
| Magnetic Laplacian Encoder | Freq. | 10 minimum eigv. |
| | Use attention | No |
| | FFN layers | 2 |

Table 9: Hyperparameters used for Dir-GNN models.

where $y_i$ is the ground truth label for reaction $i$ and $\hat{y}_i$ is the predicted label for reaction $i$. Since the dataset is highly imbalanced, we used the weighted version of the loss function, where the weight of the positive class is the inverse of the proportion of positive samples in the dataset.

# G METABOLIC NETWORK GRAPH CONSTRUCTION

In this Section, we explain how the metabolic network is transformed into a graph and how node features are extracted. The general pipeline for the transformation is shown in Figure 10. We will use this figure to exemplify the process and the design decisions taken. We start from the metabolic network shown on the left of the figure. This network is composed of 3 reactions and 2 metabolites. The first challenge that arises is the fact that certain metabolic reactions are considered reversible. This means that these reactions can be performed in both directions. We therefore need to model this fact. One naive solution is to simply link all input and output metabolites of a reaction with undirected edges. However, this solution removes the 2 sets of consumed and produced metabolites. Another solution is to label the edges of each side of the reaction (e.g. use a one-hot encoding on the edges of one side of the reaction [0, 1] and the opposite encoding on the other side [1, 0]). However, this unintentionally creates two separate sets of metabolites which might not be desirable. To solve this problem, we resorted to the approach used with LPPs when dealing with metabolic networks Ebrahim et al. (2013), and created two separate reactions for each reversible reaction. For instance, if we assume that reaction $r_2$ is reversible, we create two reactions with opposite direction $r_2^{\leftarrow}$ and $r_2^{\rightarrow}$, as shown in the first step of the figure.

Once the reversible reactions are resolved, to compute the positional encodings we need to transform the bipartite graph into a unipartite graph. To achieve this, we just use a one-hot encoding to

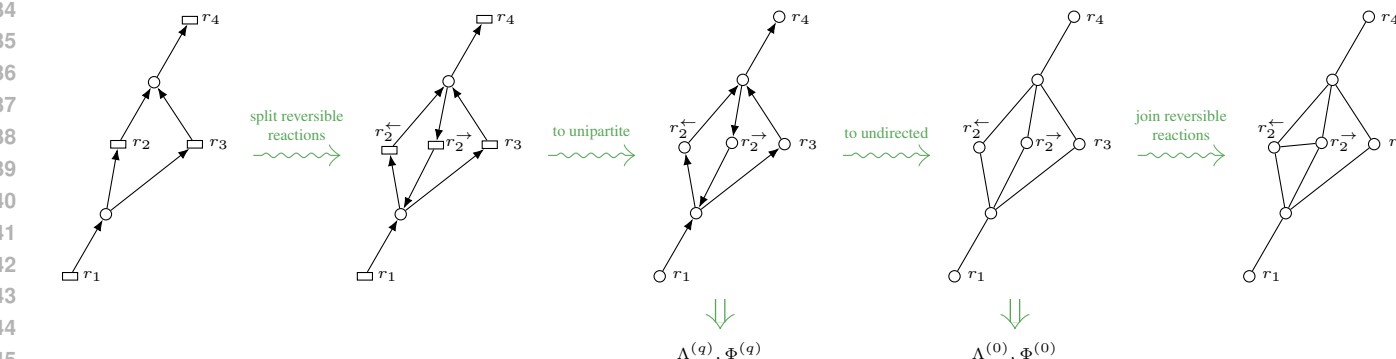

Figure 10: Metabolic network transformation. The Figure shows the steps undertaken to transform a metabolic network into a graph and how the spectral features are computed.

differentiate between reactions and metabolites and directly treat the graph as unipartite. This is shown in the third graph of the figure. On this unipartite graph, we are now able to compute the eigenvalues and eigenvectors of the magnetic Laplacian as explained in the paper.

In the cases when it is desired to also compute the combinatorial Laplacian of the graph, we simply ignore the direction of the edges and treat the graph as undirected. This results in the fourth graph of the figure.

In addition to the positional encodings and the node features aforementioned, to provide more information to the model, we also include a flag in each node features indicating whether the node is the objective reaction, and another flag indicating whether the node is a source or sink reaction in the network.

With the procedure exposed until now, empirically it usually happened that the two reactions created from a reversible reaction had different labels. This is, if $r_2$ is reversible, then $r_2^{\leftarrow}$ and $r_2^{\rightarrow}$ could be classified differently. This is an issue since reactions $r_2^{\leftarrow}$ and $r_2^{\rightarrow}$ are the same reaction, just with different directions. To solve this problem, we decided to add an extra edge between the two reactions created from a reversible reaction. This helps to propagate the information during message passing and seems to solve the problem. This is shown in the fifth graph of the figure. To differentiate between the previous edges of the network, and the new edges created, we used a one-hot encoding and included it as edge features.

# H  PERFORMANCE COMPARISON

Our proposed approach enables the use of GPUs to identify essential reactions in GEMs, which was not possible before. To show this, we used the model MODEL1011090001 from the Biomodels database Malik-Sheriff et al. (2020) which contains 3393 reactions and 2572 metabolites. The wall clock time, using the GPS + LapPE + Dir-Gated model described in Table 1, was of 0.09248 seconds with a GPU NVIDIA 3090 RTX with 24GB of memory. The wall clock time solving the FBA LPPs with GLPK solver was 6.7725 seconds with CPU Intel Core i5-9300H CPU @ 2.40GHz x 8

