# OpenReview forum: "Directed graph transformers meet metabolic networks"
_ICLR.cc/2025/Conference — Submitted to ICLR 2025_

### Official Review · Reviewer_T5V7 · 2024-11-03

**Soundness:** 4
**Presentation:** 3
**Contribution:** 2
**Rating:** 6
**Confidence:** 2

**Summary:**

The paper introduces metabolic transformers, a new transformer based architecture tailored for processing metabolic networks. They apply this model to prediction of reaction essentiality in metabolic networks.

**Strengths:**

The paper will release a curated dataset of >20k GEMs from more than 100 different microorganisms.

Introduces "metabolic transformers"

**Weaknesses:**

The problem of predicting reaction essentiality is not adequately motivated, given there are already other efficient means of solving it (LP). In comparison to this, the authors claim that their metabolic transformer can process all reactions in a network in parallel, but it is not clear in what context could this be necessary.

**Questions:**

As the authors mention, reaction essentiality can be computed by solving an LP program for each reaction. Can the authors further comment on the advantages of their metabolic transformer approach vs. the LP based one ?

Can the metabolic transformer be used to address other more challenging problems about metabolic networks ? For example, sampling feasible flux vectors, solving convex optimization problems posed on metabolic networks (see arXiv:1501.02454) ?

---

> ### Author Response · Authors · 2024-11-25
> **Review T5V7 Response**
>
> We thank the reviewer for evaluating our work and providing insightful comments. Below we address the concerns raised by the reviewer:
> ___
>
> **Weakness 1 Answer:**
>
> Thank you. It is true that we have not been clear enough with the motivation for our work. To clarify this, we have added the following line in the manuscript:
>
> “In this work, a model is proposed that is able to generalise essentiality reaction prediction across different microorganisms by pre-training on synthetic knock-out data.
> It is shown that the model can serve for effective transfer learning to real metabolic data, in this case, gene essentiality obtained experimentally in the lab.
> This is in contrast with existing LP methods that cannot generalize outside the modelled problem.
> To the best of our knowledge, this is the first model to successfully demonstrate transfer learning on metabolic data.”
>
> In summary, we don't want to make essential computation more efficient, but rather to enable transfer learning on metabolism, motivated by the scarcity of the data and the difficulty in modelling complex phenomena. Regarding processing reactions in parallel, this is just to show computationally the complexity that our problem involves. With LP-based methods, essential computation involves individual knockout over all reactions (solving one LP for each reaction), while here it is performed through a single-pass of the model. While one approach is not necessarily more efficient than the other, it gives a good insight into the complexity of the problem.
>
> ___
>
> **Question 1 Answer:**
>
> We believe that part of the question was answered in the previous question.
>
> Regarding the advantages of our method against LP-based, **our method enables data-driven prediction and transfer learning without explicitly modelling the biological phenomena (which is indeed impossible to perform with LP-based optimization).**
>
>
> Regarding the application to other metabolic network problems, we believe that the metabolic transformer trained with our objective should potentially capture enough information about the metabolic network to serve as a useful backbone for other different tasks. Our work shows good performance approximating essential reactions (which is indeed a convex optimization problem). Other works have proven theoretically that GNNs are powerful enough to approximate convex optimization solutions (see [1]), however, these are bounded by the number of message-passing layers. From our work,  we believe that:
>
> - The metabolic transformer can potentially serve as a backbone for transfer learning on other metabolic-related tasks.
> - From our results, we believe that the architecture GPS+LapPE+Dir-Gated could serve as an adequate architecture also for other metabolic network modelling tasks.
>
> Regarding sampling feasible flux vectors, we recall that LP can have from zero to an infinite number of
> solutions. Therefore, a deterministic model is not able to sample different fluxes. We believe, however, that the model studied here can help towards this goal, nevertheless, different training and inference schemes are needed for such task. We thank the reviewer for their question.
>
> References:
>
> - [1] Qian et al. Exploring the Power of Graph Neural Networks in Solving Linear Optimization Problems
> ___
>
> We hope we have addressed clearly the concerns raised by the reviewer and that we have helped in the clarification and evaluation of the manuscript. We are open for further discussion if required.

---

> > ### Comment · Reviewer_T5V7 · 2024-11-26
> >
> > Thank you for the answer. I have updated my score.

---

### Official Review · Reviewer_t5ed · 2024-11-04

**Soundness:** 2
**Presentation:** 2
**Contribution:** 2
**Rating:** 3
**Confidence:** 3

**Summary:**

The paper presents a novel graph transformer architecture designed for holistic identification of essential reactions in genome-scale models (GEMs), leveraging a large dataset of over 23,000 GEMs from a variety of microorganisms. The authors formally define "essential reactions" and employ synthetic knock-out data for training, which they claim addresses the computational limitations of current methods in handling large-scale metabolic data.

**Strengths:**

1. Addressing the identification of essential reactions in metabolic networks is a significant
2. The proposed dataset seems to be new for this direction

**Weaknesses:**

1. The most insightful section to formulate the essential reaction prediction problem is not this paper's contribution, as it's already formally stated in FlowGAT's Method

2. The major issue of this paper is a simple application of directed graph transformer on essential reaction prediction task. It lacks novelty in the methodological approach; the architectural innovations appear incremental, as the "directed graph transformer" already exists and is not proposed by authors

3. the experimental results are weak. Because the validation of the synthetic data and the incongruent results between F1 scores and ROC AUC metrics suggest potential issues in model evaluation and performance stability.

**Questions:**

1. Is there any novelty in the model architecture? what's the difference between this architecture and "Transformers Meet Directed Graphs" https://arxiv.org/pdf/2302.00049

2. In Sec. 5, "Metabolic Transformer" refers to the GPS-based or SAT-based model?

3. In Tab. 1, why does FlowGAT get a high F1 while ROC AUC < 0.5 indicates non-learning?

4. Is FlowGAT the only baseline? How about other standard GNNs or Graph Transformers?

5. Why in Tab. 2, does FlowGAT have much higher F1 while much lower ROC AUC?

---

> ### Author Response · Authors · 2024-11-25
> **Review t5ed Response**
>
> We thank the reviewer for evaluating our work and providing insightful comments. Below we address the concerns raised by the reviewer:
> ___
>
> **Weakness 1 Answer:** Thank you for identifying this weakness. We are sorry that this was not clear enough. There is a substantial difference between FlowGAT method and ours. **FlowGAT is an ad-hoc method optimized particularly for transductive learning on only 1 microorganism**, with around 50 test samples. **Our method is the first contribution to generalize essential prediction among 100s of microorganisms with 1000s of samples for each node in an inductive manner**. We believe that this enables the first step towards the construction of large-pretrained metabolic network models.
>
> ___
>
> **Weakness 2 Answer:** Thank you for remarking on this. It is true that works exist that propose directed graph transformers. However, as far as we know, this is the first work that combines the directional message-passing from [2] with existing graph transformer architectures. Additionally, **this work also shows experimentally, that undirected positional encodings + directed message passing exceeds directional positional encodings**, which seems at first as a totally counterintuitive idea. This is a very valuable insight that we have not seen in any other work.
>
> In addition to this, we want to note that, **the ultimate goal of this work is not to propose a novel “directed graph transformer”, but to answer the question “What is the best neural network model for metabolic modelling?”**. Our claims that the proposed model is the best one for our case are supported both theoretically and experimentally
> in our work.
>
> Refernces:
>
> - [1] Rossi et al. Edge directionality improves learning on heterophilic graphs, 2024.
>
> ___
>
> **Weakness 3 Answer:** Thank you for noting this which is indeed interesting. There is however no issue with the model evaluation. The values reported are correct. We answer to this in the questions below.
>
> ___
>
> **Question 1 Answer:** The architecture shown in “Transformers Meet Directed Graphs” is the equivalent of SAT+MagLapPE+GCN in our work. In our work, besides that, we also investigated, the GPS architecture (not only the SAT), we studied both laplacian (LapPE) and magnetic laplacian (MagLapPE) as positional encoding (not only MagLapPE), and we included directional message passing (DirGCN, DirGAT, etc) in the graph transformer. This differs with the work from “Transformers Meet Directed Graphs”, in that directionality information is provided in message-passing, not only in positional encodings.
>
> ___
>
> **Question 2 Answer:** Thank you for your question. In Sec 5, we used the best-performing model according to the results from Table 1, this is, GPS+LapPE+Dir-Gated. We have included a clarification sentence in the paper to indicate this:
>
> “As metabolic transformer, here we used the best-performing model according to Table 1, this is, GPS+LapPE+Dir-Gated.”
>
> ___
>
> **Question 3 Answer:**
>
> We have further investigated this. To provide an insight into this, we have obtained this confusion matrices from the best F1-score models in each case. Below we show the confusion matrix from FlowGAT on synthetic reaction essentiality:
>
> |           	| Predicted Positive | Predicted Negative |
> |---------------|---------------------|---------------------|
> | **Actual Positive** | (TP): 37  | (FN): 0 |
> | **Actual Negative** | (FP): 14 | (TN): 0  |
>
> For comparison, here we include the confusion matrix from the metabolic transformer on the same task:
>
> |           	| Predicted Positive | Predicted Negative |
> |---------------|---------------------|---------------------|
> | **Actual Positive** | (TP): 27 | (FN): 3 |
> | **Actual Negative** | (FP): 11 | (TN): 10  |
>
> From this, we can see that **FlowGAT (almost) always predicts samples as positive class. Since this is the majority class the F1-score obtained is considerably larger**. However, **the discriminative power of FlowGAT is very poor regarding the negative class, which is why the ROC AUC shows indeed non-learning**. Our model, on the other hand, has a lower rate of True Positives, but the performance with the negative class is more balanced which explains the lower F1-score but larger ROC AUC.
>
> To further clarify this in our work, we have included confusion matrices for our experiments in Figures 6 and 7, and we have repeated the experiments computing the opposite problem, this is, predicting non-essentiality (the minority class instead of the majority class). The results, reported in Tables 2 and 3 show a substantial difference
> between our model's performance and FlowGAT.

---

> ### Author Response · Authors · 2024-11-25
> **Review t5ed Response**
>
> **Question 4 Answer:** We agree that more baselines can be provided with non-dedicated models. However, the ablation studies in Appendix D, and theoretical results from Appendix B already show that message-passing alone or undirected graph transformers are limited in their application to metabolic modelling tasks. Moreover, the work from FlowGAT already showed that their model exceeded the performance from GAT.
> ___
>
> **Question 5 Answer:** Thank you, an answer is provided to this in Question 3
> ___
>
> We hope we have adressed clearly the concerns raised by the reviewer and that we have helped in the clarification and evaluation of the manuscript. We are open for further discussion if required.

---

> > ### Author Response · Authors · 2024-12-02
> > **Requesting to check our last response**
> >
> > Dear reviewer,
> >
> > Thank you once again for your insightful feedback on our work. We have carefully reviewed and addressed all your concerns in the revised manuscript. As today is the final day for submitting rebuttals, we hope you have had the opportunity to review our responses to your comments.
> > We kindly ask you if you could provide any last thoughts or, if you find our responses are satisfactory, to consider revising your score for our submission.
> >
> > Your final input would be invaluable in ensuring a comprehensive review process.
> > Thank you for your time and thoughtful consideration.

---

### Official Review · Reviewer_Csnq · 2024-11-04

**Soundness:** 2
**Presentation:** 2
**Contribution:** 2
**Rating:** 5
**Confidence:** 4

**Summary:**

The paper is aimed at predicting gene essentiality in metabolic networks. The authors propose "Metabolic Transformer", a directed-graph-based transformer architecture for learning the task.

**Strengths:**

The paper proposes a synergistic simulation-machine learning approach to an important problem in metabolic engineering, by combining Flux Balance Analysis, a linear programming-based approach for simulating the steady-state of a metabolic network, with a graph Transformer architecture. The Transformer part is based on recent approaches (Chen et al., 2022, and Rampasek et al, 2022).

**Weaknesses:**

As the authors mention, the Transformer architecture is based on prior work, and not novel in itself. The authors claim that a machine learning model trained on the results from FBA can provide the ability to scale essentiality prediction beyond what is possible with FBA, but do not provide a convincing example in the experimental validation.

The experimental results also do not show conclusively that the method performs better than prior work, the F1 scores are similar or lower, while the AUROC is higher - however, information about precision-recall curve (AUPR) is not provided.

The authors may want to reconsider the use of the work "synthetic", it seems from that manuscript that it is used as equivalent to "in silico". "Synthetic" is typically used as part of the term "synthetic lethality" in gene essentiality studies.

**Questions:**

What are the AUPR values in the experiments?

---

> ### Author Response · Authors · 2024-11-25
> **Review Csnq Response**
>
> We thank the reviewer for evaluating our work and providing insightful comments.
> Below we address the concerns raised by the reviewer:
>
> ___
>
> **Weakness 1 Answer**: While it is true that the base Transformer used in this work is based on prior works (Chen et al., 2022, and Rampasek et al, 2022, as well noted by the reviewer), it is also important to mention that both of those approaches are based on results on *undirected graphs*. While some works have been proposed to address directed graphs, see [1, 2], there is no agreement in literature on how to model directed graphs, as both approaches are completely different:
>
> - [1] Introduces directionality information with magnetic laplacian.
> - [2] Introduces directionality in message-passing.
>
> Since no clear agreement exists on GNN applied to directed graphs, we had to conduct our reported experimentations, combining different message-passing schemes, positional encodings and transformers architectures. We argue that the combination of undirected-graph positional encoding + directed-message passing, which shows the best results in our work, **is indeed a novel contribution**, since no previous work exists where such architecture is used. We believe that this insight would be especially useful for the graph-learning community and would help towards scaling graph transformers in directed datasets.
>
> Additionally, we want to highlight that **the goal of this work is not to contribute with a novel graph transformer architecture**, but **to answer the question: "what is the best architecture for modelling metabolic networks?"**. Through the work, we provide theoretical and experimental results to base our claims towards this goal.
>
> References:
>
> - [1] Geisler et al., Transformers Meet Directed Graphs, 2023.
> - [2] Rossi et al. Edge directionality improves learning on heterophilic graphs, 2024.
>
> ___
>
> **Weakness 2 Answer:** Thank you for your comment. We are sorry that the contribution of the work is not clear enough. One of the results presented in this work is the finetuning of gene essentiality from a model pre-trained on reaction essentiality. In other words, **this is the first work that shows successful transfer learning on metabolic networks.** The experimental validation provided is from the data from Escherichia coli. While we would like to extend our experimental results, metabolic data is very scarce and holistic (in vitro) essentiality information is only available at the moment for Escherichia coli.
>
> ___
>
> **Weakness 3 Answer:** Thank you for pointing out that the F1 and ROC results are inconclusive. We have further investigated this. To clarify this matter, below we show the confusion matrix from FlowGAT on synthetic reaction essentiality:
>
> |           	| Predicted Positive | Predicted Negative |
> |---------------|---------------------|---------------------|
> | **Actual Positive** | (TP): 37  | (FN): 0 |
> | **Actual Negative** | (FP): 14 | (TN): 0  |
>
> For comparison, here we include the confusion matrix from the metabolic transformer on the same task:
>
> |           	| Predicted Positive | Predicted Negative |
> |---------------|---------------------|---------------------|
> | **Actual Positive** | (TP): 27 | (FN): 3 |
> | **Actual Negative** | (FP): 11 | (TN): 10  |
>
> From the above matrices, we can see that **FlowGAT (almost) always classifies all reactions as essential**. Since **the positive class is the majority class here, the F1 score obtained is high**. However, the confusion matrix shows that indeed the discriminative power regarding the negative class is very poor (or nonexistent). This result is in agreement with the AUC ROC conclusion, this is, **that FlowGAT has poor discriminative performance**.
>
> To further clarify this, we have run additional experiments, reported in Tables 2 and 3 in the updated manuscript, where we compute the opposite problem, this is, predicting non-essentiality (the minority class). These further results strongly support that FlowGAT shows very poor discriminative performance and that our model performs considerably better.
>
> See for instance, the Table 2, where (+) denotes predicting the positive class (essentiality), and (-) denoted predicting the negative class (non-essentiality):
>
> | Model                                      | F1 (+) ↑            | ROC AUC (+) ↑        | F1 (-) ↑            | ROC AUC (-) ↑        |
> |--------------------------------------------|---------------------|----------------------|---------------------|----------------------|
> | FlowGAT [hasibi2024]            | **0.85 ± 0.033**    | 0.495 ± 0.053        | 0.02 ± 0.04         | 0.525 ± 0.075        |
> | Metabolic Transformer (ours)               | 0.845 ± 0.027       | **0.857 ± 0.022**    | **0.69 ± 0.026**    | **0.849 ± 0.017**    |

---

> ### Author Response · Authors · 2024-11-25
> **Review Csnq Response**
>
> **Question 1 Answer:** Here we provide the PRAUC values.
>
> The table below contains the results from "real essential gene prediction" as explained in the previous comment, and as denoted in the paper:
>
> | Model                                  | PRAUC (+)           | PRAUC (-)           |
> |----------------------------------------|---------------------|---------------------|
> | FlowGAT (Hasibi et al. (2024))         | 0.805 ± 0.053       | 0.432 ± 0.137       |
> | Metabolic Transformer (ours)       | **0.933 ± 0.0369**  | **0.608 ± 0.107**   |
>
> The table below contains results on "synthetic essential reactions" prediction with similar format:
>
> | Model                                  | PRAUC (+)           | PRAUC (-)           |
> |----------------------------------------|---------------------|---------------------|
> | FlowGAT (Hasibi et al. (2024))         | 0.740 ± 0.060       | 0.351 ± 0.064       |
> | Metabolic Transformer (ours)       | **0.765 ± 0.029**   | **0.766 ± 0.067**   |
>
> We believe that both tables are in agreement with the results from F1 and AUC ROC and the discussion provided in the previous answer, this is, that the metabolic transformer discriminative performance is considerably better for both problems.
>
> ___
>
> We hope we have adressed clearly the concerns raised by the reviewer and that we have helped in the clarification and evaluation of the manuscript. We are open for further discussion if required.

---

> > ### Author Response · Authors · 2024-12-02
> > **Requesting to check our last response**
> >
> > Dear reviewer,
> >
> > Thank you once again for your insightful feedback on our work. We have carefully reviewed and addressed all your concerns in the revised manuscript. As today is the final day for submitting rebuttals, we hope you have had the opportunity to review our responses to your comments.
> > We kindly ask you if you could provide any last thoughts or, if you find our responses are satisfactory, to consider revising your score for our submission.
> >
> > Your final input would be invaluable in ensuring a comprehensive review process.
> > Thank you for your time and thoughtful consideration.

---

> > > ### Comment · Reviewer_Csnq · 2024-12-03
> > >
> > > Thank you for the additional information and results. I have updated my score to 5; I believe the submission falls somewhat below the threshold on the novelty axis.

---

### Meta-Review · Area_Chair_EeqE · 2024-12-16

**Metareview:**

This paper is an interesting development of using Graph transfomers for understanding Metabolic networks. The main technical novelty in the space of Graph transformers is the use of Directed graphs, in contrast to previous work which was in the space of Undirected graphs.
While there is definitely praiseworthy contributions but the referees were not convinced that this is a strong enough contribution to a conference like ICLR.

**Additional Comments On Reviewer Discussion:**

In the discussion during the rebuttal the reviewers, though acknowledged the efforts of the author(s), could not make a strong  recommendation for accept, unfortunately. Hopefully if the paper can address the concerns raised by the reviewers, specially the experimental part, it can be a strong submission.

---

### Decision · Program_Chairs · 2025-01-22

Reject